# Automatic Recognition of High-Density Epileptic EEG Using Support Vector Machine and Gradient-Boosting Decision Tree

**DOI:** 10.3390/brainsci12091197

**Published:** 2022-09-05

**Authors:** Jiaxiu He, Li Yang, Ding Liu, Zhi Song

**Affiliations:** Department of Neurology, The Third Xiangya Hospital of CSU, Tongzipo Street, Changsha 410013, China

**Keywords:** EEG, epilepsy, machine learning, SVM, GBDT

## Abstract

Background: Epilepsy (Ep) is a chronic neural disease. The diagnosis of epilepsy depends on detailed seizure history and scalp electroencephalogram (EEG) examinations. The automatic recognition of epileptic EEG is an artificial intelligence application developed from machine learning (ML). Purpose: This study compares the classification effects of two kinds of classifiers by controlling the EEG data source and characteristic values. Method: All EEG data were collected by GSN HydroCel 256 leads and high-density EEG from Xiangya Third Hospital. This study used time-domain features (mean, kurtosis and skewness processed by empirical mode decomposition (EMD) and three IMFs), a frequency-domain feature (power spectrum density, PSD) and a non-linear feature (Shannon entropy). Support vector machine (SVM) and gradient-boosting decision tree (GBDT) classifiers were used to recognize epileptic EEG. Result: The result of the SVM classifier showed an accuracy of 72.00%, precision of 73.98%, and an F1_score of 82.28%. Meanwhile, the result of the GBDT classifier showed a sensitivity of 98.57%, precision of 89.13%, F1_score of 93.40%, and an AUC of 0.9119. Conclusion: The comparison of GBDT and SVM by controlling the variables of the feature values and parameters of a classifier is presented. GBDT obtained the better classification accuracy (90.00%) and F1_score (93.40%).

## 1. Introduction

Epilepsy is a chronic neural disease. According to the International League Against Epilepsy (ILAE), a seizure does not necessarily mean that a person has epilepsy, unless criteria for the diagnosis of epilepsy are met. The diagnosis of epilepsy depends on many factors, such as detailed and accurate seizure history and some assistant examinations, particularly electroencephalograms (EEGs), including normal EEGs, video EEGs (VEEGs) and ambulatory EEGs (AEEGs) [1]. The EEG is a powerful and important tool for the diagnosis and classification of seizure and epilepsy [2,3]. Interictal epileptic EEG is essential for the diagnosis of epilepsy, as epileptic EEG features may be obscured by artifacts [4].

Epilepsy (Ep) is a chronic neural disease, with recurrent, persistent and episodic characteristics. There are about five million epilepsy patients in the world, and there are about two million new cases every year, with an incidence of about 0.7% [5]. Epilepsy is caused by the hyper-synchronous discharge of neurons, which is an abnormal state, accompanied by the formation of abnormal epileptic brain networks. At this time, neurons show an extremely active discharge activity, which leads to a series of seizure symptoms such as fall down, fracture, coma, and so on. Almost all forms of epilepsy can be controlled by drug therapy. Thus, early diagnosis is the most important step in the treatment of epilepsy.

As we all know, scalp EEG is a significant auxiliary examination in the diagnosis of epilepsy. Electroencephalogram (EEG) is a wave image that records spontaneous bioelectricity in the brain and amplifies it through electrode leads. EEG, which is non-invasive and simple, is chosen as the first diagnostic method. During clinical work, VEEG and AEEG are commonly used in the diagnosis of epilepsy because long-time monitoring can increase the chance of detecting seizures. However, this method requires lots of time and energy to recognize epileptic EEG, and can easily lead to manual errors. Artificial intelligence may help solve this difficulty. With the development of technology, the number of electrodes used in clinical EEG examination ranges from 16 electrodes to 256 electrodes; the more detail that can be observed and the greater the amount of data obtained, the greater chance physicians have at diagnosis [6]. The automatic recognition of epileptic EEG is an artificial intelligence application developed from machine learning (ML).

The automatic recognition of epileptic EEG is an artificial intelligence application developed from machine learning (ML). Considering that the automatic recognition of epileptic EEG is an algorithm to distinguish epileptic EEG from non-epileptic EEG, the algorithm is the binary classification, and the result is “yes or no”. Slevakumari et al. [7] obtained a sensitivity of 95.75%, specificity of 96.55%, and accuracy of 95.63% though SVM to distinguish epileptic EEG; Rizal et al. [8] used SVM classification to obtain accuracy result of 97.70%; and Jaiswal et al. [9] obtained an accuracy of 97.50% through SVM classification. Li et al. [10] obtained a sensitivity of 95.50%, a specificity of 98.00%, and an accuracy of 94.00% by the random forest method. Hu et al. [11] obtained a classification result with the highest accuracy of 92.0% using GBDT classification to distinguish EEG databases; another study [12] in 2019 obtained an accuracy of 84.22% with GBDT classification. Different feature values can affect the classification results. This makes it impossible to directly compare the classifier effects by classification results.

In this paper, we compare different EEG classifiers based on the same clinical high-lead EEG data and the same feature values to find a more suitable EGG classifier for epilepsy.

## 2. Method

The experiment is an EEG classification experiment. This study used the same characteristic values in order to compare the classification effects of different classifiers. The technology roadmap is shown in Figure 1.

### 2.1. EEG Data

This study included 21 participants, with 15 epileptic patients and 6 healthy participants, and a total of 105 EEG data. All participants in this study were from the Department of Neurology of Xiangya Third Hospital. The inclusion and exclusion criteria were as follows: Inclusion criteria: (1) diagnosis obeying the epilepsy diagnosis standard of the International League Against Epilepsy (ILAE); (2) age ≥ 15 years. Exclusion criteria: (1) a history of other brain-related diseases (trauma, infection, and so on); (2) unable to complete EEG tasks independently; (3) cannot tolerate long-term EEG examination.

All EEG data were collected by GSN HydroCel 256 leads and high-density EEG (EGI company, from Shanghai Nohe Medical Company, LTD, Shanghai, China). Then, we completed the pre-treatment of EEG, including filtering and ICA, with the EEGLAB toolbox [13] (2021.1, Arnaud Delorme and Scott Makeig, CA, USA) and Matlab software (2017b, MathWorks Company, Natick, MA, USA). In this study, there were 105,256 lead EEG data lasting 60 s, with the frequency band of 0–80 Hz.

### 2.2. Feature Extraction

This study used PSD, Shannon entropy, mean, kurtosis and skewness as characteristic values, and mean, kurtosis and skewness were processed by EMD.

#### 2.2.1. Power Spectral Density (PSD)

PSD, known as the power spectrum, represents the signal power within a unit frequency band. The PSD shows the changes in signal power by frequency, that is, the power distribution of the signal in the frequency domain. The basic definition of PSD can be expressed as:(1)P=1T∫−T2T2f2(t)dt

In Equation (1), *P* represents the average power of power signal f(t) over the time period [−T2, T2]. Additionally, the unit of PSD is $V^{2}/Hz$. In order to reduce the bias during PSD analysis, Pwelch’s method [14] was used in the experiment.

#### 2.2.2. Shannon Entropy

Shannon entropy, also known as Information entropy, was proposed by Clause Shannon [15] in his paper “Mathematical Principles of Communication” in 1948. Shannon pointed out that information is used to eliminate random uncertainties. The definition of Shannon entropy [16] is:(2)$$H\left(X\right)=-\sum p_{i}^{\left(n\right)}\log p_{i}^{\left(n\right)}$$

In Equation (2), H(X) represents the sum of the probability of *n* events, and each probability of each event is $p_{1},p_{2},\cdots ,p_{n}$. Additionally, 0log0=0, where p(x) is the probability of the event. The unit of Shannon extropy is bits. Shannon entropy can be used to describe the complexity of a system. The more complex a system is, the more different kinds of situations may occur, and the bigger the Shannon entropy of the system is. The simpler a system is, the fewer different kinds of situations may occur, and the smaller the Shannon entropy of the system is, which can be zero if it is simple enough.

#### 2.2.3. Empirical Mode Decomposition (EMD)

Empirical mode decomposition is a new signal processing method creatively proposed by Huang E in NASA [17]. EMD can transform non-stationary signals into stationary signals to obtain more accurate EEG signals. The key point of this signal processing is that through the mode decomposition algorithm, complex signals can be decomposed into intrinsic mode function (IMF). The EMD transforms the non-stationary signals into stationary signals, making the instantaneous signals meaningful.

In this experiment, we used the EMD method to obtain three IMFs from each channel. Then, we calculated the time-domain values of the EEG signals through the IMFs. Mean, kurtosis and skewness were used to value the EEG characteristics.

The calculation methods of mean, kurtosis and skewness are shown as follows:(3)$$Mean\left(X\right)=\mu \left(x\right)=\underset{N\to \infty }{\lim}\frac{1}{N}x_{n}\left(t\right)$$
(4)$$Skew\left(X\right)=E\left[{\left(\frac{X-\mu }{\sigma }\right)}^{3}\right]$$
(5)$$Kurt\left(X\right)=E\left[{\left(\frac{X-\mu }{\sigma }\right)}^{4}\right]=\frac{E\left[{\left(X-\mu \right)}^{4}\right]}{{\left(E\left[{\left(X-\mu \right)}^{2}\right]\right)}^{2}}$$

In Equation (3), X(t) represents the signal in the time domain, and *N* is the sampling point in the calculation. When the sampling points are infinitely many, as N→∞, we obtain the mean of the whole signal by (3). In Equations (4) and (5), X(t) also represents the signal in the time domain, μ is the mean of the time signal, and σ is the standard deviation of the time signal in which $\sigma =\sqrt{\frac{1}{N}\sum_{n=1}^{N}\left(x{\left(t\right)}_{n}-\mu_{x}\right)}$. 

### 2.3. Classifier

Our study used support vector machine (SVM) and gradient-boosting decision tree (GBDT) classifiers to distinguish epileptic EEG from non-epileptic EEG.

#### 2.3.1. Support Vector Machine (SVM)

SVM is a binary classification model. It can be divided into linear models and nonlinear models according to the type of input data [18]. In EEG classification, linear-separable SVM is more commonly used. In our experiment, the EEG data were divided into a training set and testing set with a ratio of 7:3 by random stratified sampling. Then, we selected the characteristic value by normalization and trained the SVM classifier by the RBF kernel [19] method. This method employed the successive grid search technique to find the optimal model parameter values C and gamma: the optimization range of C was $2^{j}$, which *j* traversed from −4 to 4 in steps of 1; the optimization range of gamma was $2^{i}$, which *i* traversed from −4 to 4 in steps of 1. Then, we obtained the best classifier model by 5-flod cross-validation. Finally, we obtained the classification results from the best model. The SVM classification process is shown in Figure 2.

#### 2.3.2. GBDT Classifier

The gradient-boosting decision tree (GBDT) is a boosting algorithm based on the decision tree proposed by Firedman [20] in 2001. The GBDT algorithm uses a gradient algorithm, reducing the over-fitting problems of the traditional decision tree and making the classification more accurate and precise. Commonly, classification and regression tree (CART) is a kind of weak classifier in iterative classification; in each iteration classification, each weak classifier is trained based on the previous one fitted by the gradient algorithm [21]. In our study, the EEG database was divided into a training set and testing set with a ratio of 8:2 by random stratified sampling. Then, we selected characteristic values by t-test and normalization. Then, we obtained the best GBDT classifier model by 5-flod. Finally, we obtained the classification results from the best model. The GBDT classification process is shown as Figure 3.

### 2.4. Statistical Evaluation

The experimental results are a dichotomous result, because the classification shows two kinds of results which are “the EEG is epileptic EEG” and “the EEG is non-epileptic EEG”. The test result of “epileptic EEG sample” is the positive sample, and “non-epileptic EEG sample” is the negative sample. We used the confusion matrix to evaluate the dichotomous data in the experiment.

TP (true positive) is the number of positive samples predicted by the model; FP (false positive) is the number of negative samples predicted by the model as positive samples. FN (false negative) is the number of positive samples predicted as negative samples by the model; TN (true negative) is the number of negative samples predicted by the model as negative samples. After that, we calculate the sensitivity, specificity, accuracy, precision, and F1_score. Additionally, we draw the ROC curve and calculate the AUC value to evaluate the classifier model.
(6)$$sensitivity=recall=\frac{TN}{FP+TN}$$
(7)$$specificity=\frac{TP}{FN+TP}$$
(8)$$accuracy=\frac{TP+TN}{TN+TP+FN+FP}$$
(9)$$precision=\frac{TP}{TP+FP}$$
(10)$$F1_{score}=\frac{2\times precision\times recall}{precision+recall}$$

## 3. Result

### 3.1. Participant Information

Our study contained 21 participants, including 15 epileptic patients and 6 healthy participants. There were 6 males and 15 females in our study, as shown in Table 1. The numbers 1–15 were the people with epilepsy, and the numbers 16–21 were the healthy people. We used the nonparametric test of significance to evaluate the age of the two groups, which suggested that there was no significant difference between the age of the two groups (*p* > 0.05).

### 3.2. Classification Result

In our study, we used five kinds of characteristic values to express the information of the EEG data. Then, we classified the EEG data by two classifiers: SVM and GBDT. In order to evaluate the classifiers, we calculated sensitivity, specificity, accuracy, precision, F1_score, and AUC value, and obtained the results shown in Table 2 and Figure 4.

The result of the SVM classifier showed a sensitivity of 92.86%, specificity of 23.33%, accuracy of 72.00%, precision of 73.98%, F1_score of 82.28%, and AUC of 0.7500. Meanwhile, the result of the GBDT classifier showed a sensitivity of 98.57%, specificity of 70.00%, accuracy of 90.00%, precision of 89.13%, F1_score of 93.40%, and AUC of 0.9119. In the intuitive comparison of the results, the values of sensitivity were almost the same, but the values of specificity were very different, and the GBDT result was far better than the SVM. The comparison of accuracy, precision, AUC, and the overall evaluation index shows that the GBDT presents much better results than the SVM.

## 4. Discussion

In this paper, a comparison of GBDT and SVM by controlling the variables of the feature values and parameters of the classifier is presented. The EEG signals were acquired from 15 epileptic and health volunteers and recorded with the 256-channel GSN Hydocel. Then, the finite impulse response (FIR) filters and ICA method were applied to EEG signals for processing. The five feature values (PSD, Shannon entropy, mean, kurtosis and skewness, where mean, kurtosis and skewness were processed by EMD and three IMFs) were applied to EEG signals for describing EEG information. Finally, two classifiers (GBDT and SVM) were applied to distinguish epileptic EEG from non-epileptic EEG. GBDT obtained the better classification accuracy (90.00%) and F1_score (93.40%).

At present, SVM is still the mainstream choice in the field of EEG classification. The mainstream classifiers have good performance in epileptic EEG classification in the published studies. GBDT is a new ML classifier that is rarely applied in the recognition of epileptic EEG. Recent studies show that GBDT has a great classification performance in the classification of epileptic EEG. However, there are still no comparisons between SVM and GBDT. This study pays attention to this question and finds that GBDT, as a new classifier in the field of the automatic recognition of epileptic EEG, has a better classifier effect.

GBDT is an emerging classifier published in 2001, and is rarely used in EEG classification. Only 11 search records can be found in PubMed and Embase with the search terms “GBDT” AND “EEG”. Among these, there are only six search records from the last 3 years. In the study by Huang et al. [22] in 2021, the GBDT classifier showed a sensitivity of 85.9%, specificity of 84.0%, and accuracy of 87.4% in children’s EEG classification. SVM, as a traditional classifier, is used commonly in epileptic classification with a great performance. Zhou M [23], Wang D [24], and other teams [7,9,25,26,27,28,29] used SVM as the classifier to distinguish epileptic EEG. The details of the above classifications are shown in Table 3.

Many factors can influence the classification results. Different feature values and different classifier parameters can both affect the classification results. This makes it impossible to directly compare the classifier effects by classification results. According to Table 3, GBDT and SVM both have great performance. However, we cannot draw the conclusion of which classifier has a better classification performance.

Therefore, in order to compare the classification performance of GBDT and SVM, control variables are important. This study used the same feature values and same classifier parameters in order to control variables. Then, we used different classifiers to distinguish epileptic EEG and non-epileptic EEG. Therefore, the comparison between different classifiers makes sense.

According to the results of this study, all statistical evaluations suggest that GBDT, a rising classifier, has a better classification performance than SVM. GBDT has great sensitivity, accuracy, and F1_score in epileptic EEG recognition. Compared with the data in Table 3, the classification results of GBDT in this paper are not the best. However, as mentioned above, the selection of feature values can affect classification results, and we cannot directly evaluate a classifier by its accuracy. In this study, GBDT showed a better classification performance than SVM.

A limitation of the proposed study might be considered as the restricted number of participants. While the sample may seem small, the method shows good classification performance with an AUC of 0.9119. The recognition accuracy may be improved with more EEG signals and participants. Furthermore, feature extraction is a major step in our EEG methodology. We used time-domain features (mean, kurtosis and skew), a frequency-domain feature (PSD), and a non-linear feature (Shannon entropy) to calculate the information of EEG signals. The classification performance may be improved with the calculation of the more feature values; however, the addition of more features from 256 channels would definitely increase the complexity of the proposed method and computational burden. This makes it impossible for our method to be applied in a clinical real-time application in the future.

## 5. Conclusions

In this paper, a new automatic recognition of an epileptic EEG method comparing GBDT and SVM by controlling the variables of the feature values and parameters of the classifier was presented. After a preprocessing and feature extraction stage, this method was able to classify EEG recordings using the GBDT classifier. Our ambition is to apply this method efficiently in clinical epilepsy diagnosis to reduce the workload of physicians, increase the efficiency of epilepsy diagnosis, and benefit people with epilepsy.

## Figures and Tables

**Figure 1 brainsci-12-01197-f001:**
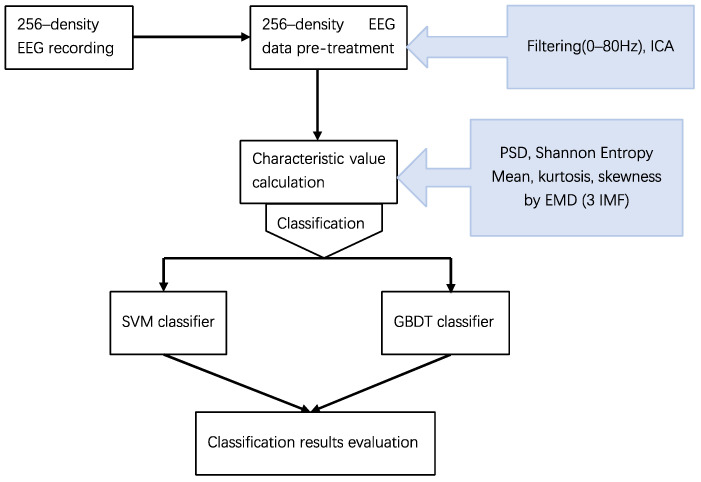
The technology roadmap of this experiment.

**Figure 2 brainsci-12-01197-f002:**
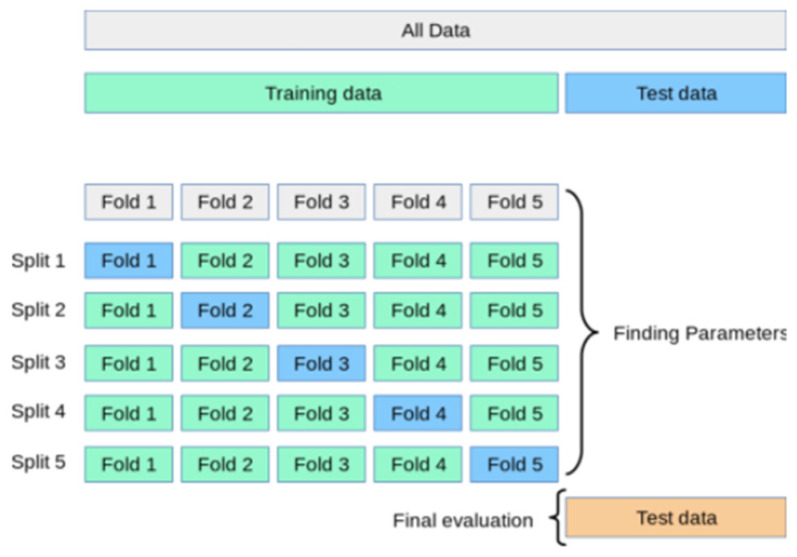
SVM classification method in this experiment. The method included 3 steps: data set division, classification model training, obtaining result from the best model.

**Figure 3 brainsci-12-01197-f003:**
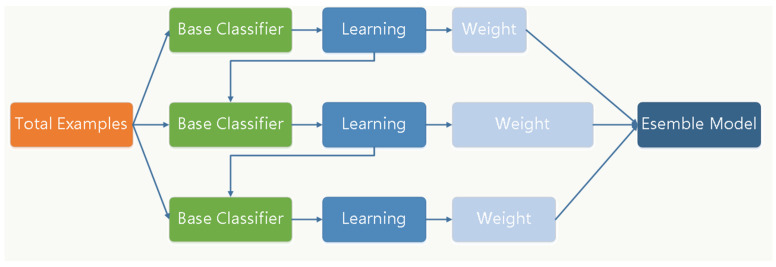
GBDT classification method in this experiment. The method included 3 steps: data set division, classification model training, and obtaining the result from the best model.

**Figure 4 brainsci-12-01197-f004:**
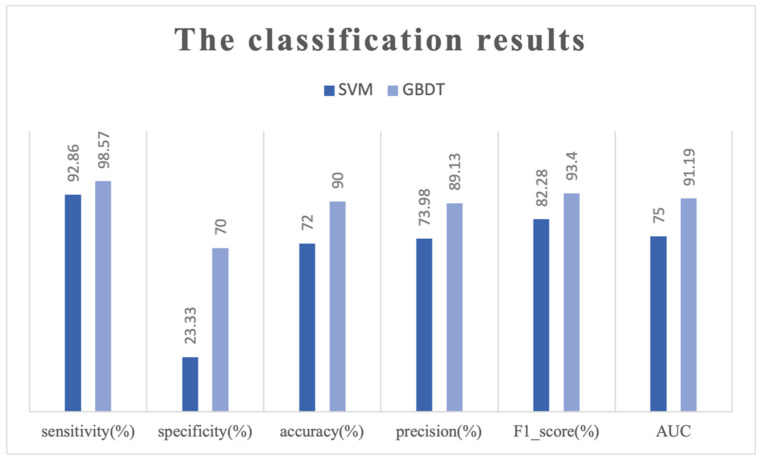
The histogram of the classification results of SVM and GBDT. Six criteria were used to evaluate the classifier model. Obviously, the classification performance of GBDT was better than SVM.

**Table 1 brainsci-12-01197-t001:** All participants’ information.

No.	Age (Yds)	Gender	Course of Disease (Yds)	Inducement	Precursor
1	26	Female	15	Pungent smell	Fear; strange taste
2	18	Male	1	Lack of sleep; tired	Headache; dizzy
3	31	Female	3	Weather change	Dizzy
4	72	Female	26	Tired; mental stimulation	Dizzy
5	31	Male	3	Weather change	Dizzy
6	17	Female	0.083	No	No
7	57	Female	5	anxiety; mood swing	No
8	26	Female	4	Tired; lack of sleep	Distraction
9	35	Male	20	Mood swing; lack of sleep	Tremble
10	31	Male	28	Fever; tired; lack of sleep	No
11	14	Female	83	No	Stomach gas rising
12	18	Female	0.083	Fever; mental stimulation	Photism; auditory hallucination
13	22	Female	6	Tired; insomnia; pregnancy	Dizzy
14	56	Female	40	Fever	Photism; auditory hallucination
15	16	Male	15	Tired; lack of sleep	Dizzy
16	21	Female	-	-	-
17	22	Female	-	-	-
18	22	Male	-	-	-
19	21	Male	-	-	-
20	22	Female	-	-	-
21	54	Female	-	-	-

**Table 2 brainsci-12-01197-t002:** The classification result of SVM and GBDT.

	Sensitivity (%)	Specificity (%)	Accuracy (%)	Precision (%)	F1_Score (%)	AUC
SVM	92.86	23.33	72.00	73.98	82.28	0.7500
GBDT	98.57	70.00	90.00	89.13	93.40	0.9119

**Table 3 brainsci-12-01197-t003:** Details of above classifications.

Author	EEG Sources	Feature Value	Method	Accuracy
Huang [22]	Unknown	Unknown	GBDT	0.846
Wang [12]	Henan Provincial People’s Hospital	Lempel–Ziv complexity; Kolmogorov complexity	GBDT	0.810
Zhou [23]	The Freiburg iEEG database	The fast Fourier transform	SVM	0.923
	CHB-MIT			0.956
Zhang [25]	Henan Provincial People’s Hospital	Sample entropy	SVM	0.900
Selvakumari [7]	CHB-MIT	Quantile, Shannon entropy, root mean square, energy et al.	SVM	0.9563
Cimbalnik [26]	Mayo Clinic	HFO features (rate); univariate features (power, amplitude, PSD); bivariate features (relative entropy, correlation)	SVM	0.839

## Data Availability

This study did not report any data.

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
