# Peer review of "Automatic Recognition of High-Density Epileptic EEG Using Support Vector Machine and Gradient-Boosting Decision Tree"

_brainsci, 2022, doi:10.3390/brainsci12091197_

Round 1

Reviewer 1 Report

The proposed method has pretty much-done people better. The advantages of this paper are the primary dataset and the GBDT method, which are rarely used in other studies. Many things need to be improved so that this paper can be accepted and published.

In the introduction, the research gap was not disclosed. Besides that, the selection of PSD, Shannon entropy, and EMD as the air extraction method was also not announced.

In the EMD process, several IMFs will be produced; it has not been explained how many IMFs have been used; there is only an explanation that five features are used, which indicates that only 1 IMF is used. Even then, there is no explanation for why only 1 IMF is used in this study.

The results section does not provide examples of normal and epileptic input signals and a comparison of the characteristics between the two.

In Table 3, only the classifier is compared, even though the feature extraction method is also very influential on accuracy.

The results only focus on the accuracy and others but do not discuss the signal characteristics, feature extraction methods, and why the classifier used can get the resulting accuracy.

Some write and data errors are also seen in the following details:

Line 77: “Mathematical Principles of Communication” in 1984, supposed to be 1948

Line 104: we got the best classifier model by 5-flod cross validation --> 5-fold cross-validation

Line 112: radio -->ratio

Line 114: 5-flod cross-validation --> 5 fold

Line 138 -139: five characteristics mean that the author used PSD, Shannon entropy, mean, skewness, and kurtosis as features. It is this means that only IMF1 from EMD used for this paper? Why?.. other methods used several IMF, for example, “Epileptic Seizure Detection in EEG Signal using EMD and Entropy (archive.org)” used 8 IMF

Author Response

Dear reviewer:

Thank you for your decision and constructive comments on my manuscript. We have carefully considered the suggestion of Reviewer and make some changes. We have tried our best to improve and made some changes in the manuscript.

The yellow part that has been revised according to your comments. Revision notes, point-to-point, are given as follows:

1. Comment: In the EMD process, several IMFs will be produced; it has not been explained how many IMFs have been used; there is only an explanation that five features are used, which indicates that only 1 IMF is used. Even then, there is no explanation for why only 1 IMF is used in this study.

1. Replay: We gratefully appreciate for your valuble suggestion. We appreciate it very much for this good suggestion, and we have done it according to your ideas. Five feature values were used in this paper, which are PSD, Shannon entropy, mean, skew and kurtosis. Among them, mean, skew and kurtosis were processed by EMD method. In this experiment, three IMF values (IMF1,IMF2,IMF3) from each channels were applied to the feature processing of three time-domain eigenvalues.

2. Comment: The results section does not provide examples of normal and epileptic input signals and a comparison of the characteristics between the two.

2.Replay: Thanks for your advice. We agree with you that feature values are important for classification. However, the aim of this paper is the comparison of two classifiers, so we  don't think it's necessary to  show all results of feature values.Maybe we can provie the files of all feature values as attachment.

3. Comment: In Table 3, only the classifier is compared, even though the feature extraction method is also very influential on accuracy.

3. Replay: Thank you for your rigorous consideration. We totally agree with you, and we added the Line226-228 in order to explain this question.

4. Comment:Some write and data errors are also seen in the following details.

4.Replay: We feel sorry for the inconvenience brought to the reviewer. We apologize for the poor language of our manuscript. We have now worked on both language and readability and have also involved native English speakers for language corrections. We really hope that the flow and language level have been substantially improved.

5. Comment:Line 138 -139: five characteristics mean that the author used PSD, Shannon entropy, mean, skewness, and kurtosis as features. It is this means that only IMF1 from EMD used for this paper? Why?.. other methods used several IMF, for example, “Epileptic Seizure Detection in EEG Signal using EMD and Entropy (archive.org)” used 8 IMF.

5. Replay: We totally understand the reviewer’s concern. We feel soory  for the inconvenience brought to the reviewer. As our first replay, we used 5 feature values in this paper. PSD and Shannon entropy were calculated directly. Mean, skew and kurtosis were calculated after EMD method. we got 3 IMFs after EMD method in each channels and used all IMFs in time-domain feature extraction.

Reviewer 2 Report

The authors present the article entitled “High density EEG classification base on SVM and GBDT”

This paper compares different EEG classifiers based on clinical high lead EEG data to find a more suitable EGG classifier for epilepsy.

The article presents the following concerns:

  • The affiliations are missing.

  • The manuscript needs hard English grammar and typing revision. Also, the title presents mistakes.

  • Avoid using first-person sentences and apostrophes.

  • The Abstract section must be restructured. This section should give a pertinent overview of the work in a single paragraph. Please check the guide for authors.

  • The citing style is out of the format.

  • Line 22-34: Are these lines correspond to the Introduction section?

  • The objective and novelty of the work are not clear. I suggest restructuring the Introduction section to identify the novelty and contributions of the work.

  • Please define each function/variable of the equations.

  • It is highly recommended to place a flow chart of the method used.

  • Subsection 2.3.1: The description of SVM is not clear. Please restructure and place a reference. Also, please justify the use of RBF kernel method.

  • Line 159: The author mentions that there are published works related to the proposed method. What is the novelty of the work?

  • In a first sight, table 3 shows greater results than the proposed method (Table 2)

  • In general, the novelty of the work is not clear. The results mentioned (Tables 2 and 3) must be described in detail because reported methods using SVM seem to perform much better than GBDT. All the manuscript must be improved.

  • line 22-34 can be justified partially with the next reference: Impact of EEG Parameters Detecting Dementia Diseases: A Systematic Review

The following misspelling should be checked:

  1. line 29: “the treatment of…” may be wordy. Consider changing by “treating…”

  2. line 36: “Automatic…” It seems that there is an article usage problem here. Add the article: “An automatic…”

  3. line 62: “cannot…” This sentence appears to be missing a verb. Consider adding a verb or rewriting the sentence by: “Cannot be” or “Cannot become”

  4. line 70: “power…” needs to add the article: “the power…” 

  5. lines 88-89: Your sentence it’s unclear or hard to follow. Consider rephrasing.

  6. line 94: It appears  that the phrase “are been” contains an unnecessary verb. Review the sentence to determine the appropriate correction: “are”, “have been” or “are begin”

  7. line 106: The phrase “kind of…” may weaken your message. Consider removing it.

  8. line 109: “precision” should be rewritten by “precise”

  9. lines 152-154: The sentence it’s hard to read. Consider rephrasing. 

  10. line 172: “the comparison of…” should be rewritten by “comparing…”

Author Response

Thank you for your decision and constructive comments on my manuscript. We have carefully considered the suggestion of Reviewer and make some changes. We have tried our best to improve and made some changes in the manuscript.

The yellow part that has been revised according to your comments. Revision notes, point-to-point, are given as follows:

1.Comment: The objective and novelty of the work are not clear. I suggest restructuring the Introduction section to identify the novelty and contributions of the work.

1.Replay: We totally understand the reviewer’s concern. Compare with other studies, our study used the primary dataset, not open-scoure EEG database. And the GBDT classifier is rarely used in other studies. We aimed to compare the classification performance between tranditional classifier(SVM) and new classifier(GBDT). We think that the advantages of this paper are the primary dataset and the GBDT method, which are rarely used in other studies.

2.Comment:

  • Line 159: The author mentions that there are published works related to the proposed method. What is the novelty of the work?

  • In a first sight, table 3 shows greater results than the proposed method (Table 2)

  • In general, the novelty of the work is not clear. The results mentioned (Tables 2 and 3) must be described in detail because reported methods using SVM seem to perform much better than GBDT. All the manuscript must be improved.

2. Replay: We gratefully thanks for the precious time the reviewer spent making constructive remarks, but we respectfully disagree with you. We believe that the comparison should be performed in the same variables. Different feature values will affect the classification results. And different parameters of classifiers will also affect the classification results. So we cannot compare directly according to Table3 and Table2. Therefore, this paper controlled the variables (feature values and classifier parameters) to cpmpare the classification performance of different classifiers.

3. Comment: Subsection 2.3.1: The description of SVM is not clear. Please restructure and place a reference. Also, please justify the use of RBF kernel method.

3.Replay: Thank you for your rigorous comments. SVM is a tranditional classifier in automatic recognition of epiletic EEG. Kernel method is a non-linear method of SVM, and RBF is a commonly mathematics method using in kernel method. We we have added the reference in the subsection 2.3.1.

4. Comment: The citing style is out of the format.

4.Replay: We are very sorry for our careless mistake and it was rectified now.

5. Comment:

  • The manuscript needs hard English grammar and typing revision. Also, the title presents mistakes.

  • Avoid using first-person sentences and apostrophes.

  • The Abstract section must be restructured. This section should give a pertinent overview of the work in a single paragraph. Please check the guide for authors.

5.Replay: We are very sorry for our careless mistake. The title now was rectified as "Automatic recognition of high-density epileptic EEG using support vector machine and gradient boosting decision tree". The abstract now was rectified and changed the structure.

6.Comment: Line 22-34 can be justified partially with the next reference: Impact of EEG Parameters Detecting Dementia Diseases: A Systematic Review.

6. Replay: Thanks for your valuable suggestion. We have cited this reference in the proper place of the revised mauscript.

7. Comment: Please define each function/variable of the equations.

7. Replay: Thanks for your valuable suggestion. We have define eachn variables of the equations as the yellow part in manuscript.

8. Comment: The following misspelling should be checked.

8. Replay: We gratefully thanks for the precious time the reviewer spent making constructive remarks. We are very sorry for our careless mistake and it was rectified now.

Reviewer 3 Report

brainsci-1847498

 High-density EEG classification base on SVM and GBDT

Author Response

Dear reviewer:

Thank you for your decision and constructive comments on my manuscript. We have carefully considered the suggestion of Reviewer and make some changes. We have tried our best to improve and made some changes in the manuscript.

The yellow part that has been revised according to your comments. Revision notes, point-to-point, are given as follows:

1.Comment: The authors want to present the contribution and novelty of this paper. 

Article innovation is low.

1.Replay: We totally understand the reviewer’s concern. In this paper, a comparison of GBDT and SVM by controlled the variables of the feature values and parameters of classifier is presented. Compare with other studies, our study used the primary dataset, not open-scoure EEG database. And the GBDT classifier is rarely used in other studies. We aimed to compare the classification performance between tranditional classifier(SVM) and new classifier(GBDT). We think that the advantages of this paper are the primary dataset and the GBDT method, which are rarely used in other studies.

2. Comment: Comparison with new references.

2. Replay: We gratefully appreciate for your valuable suggestion. It's important to emphasize that the difference between our manuscript and this references is the sourse database. This three references used the EEG databases of Bonn University and Freiburg University. The EEG data of Bonn University included resting state with eye opening, resting state with eye closing and intracranial EEG. The EEG data of Freiburg University were intracranial EEG data. In clinic work, when physician faces a person who had a seizure, the first examination for this person is scalp EEG examination, not intracranial EEG examination. Base on this situation, the EEG databases of Bonn University and Freiburg University are not suitable for diagnosisi of epilepsy. In order to reducing the workload of epilepsy diagnosis, we think that using the primary scalp EEG database is more signifacant for epilepsy diagnosis.

3. Comment: Other entropies such as: Log-Energy Entropy, Average Shannon Wavelet Entropy, Average Rényi Wavelet Entropy, Average Tsallis Wavelet Entropy, Fuzzy Entropy, should also be used to evaluate the method.

3. Replay: Thank you for your advice. We agree with you that there are many entropies which can be used in automatic recognition of epileptic EEG. However, there are many kinds of feature values in epiletic EEG classification, such as mean, skew, kurtosis, PSD, wavelet. The time-domain feature value and frequency-domain value are linear feature in order to discript EEG information. Entropy is a nonlinear index. During feature value selection, we believe that the selection should be comprehensive, not limited to the entropy. In this study, we used time-domain feature value (mean, skew, kurtosis), frequency-domain feature value (PSD) and non-linear feature value (Shannon entropy). In the further, we will try to use other entropy such as Log-Energy Entropy, Average Shannon Wavelet Entropy, Average Rényi Wavelet Entropy, Average Tsallis Wavelet Entropy, Fuzzy Entropy in the study.

4.Comment: Other criteria such as: F1-score (F1-S), and precision (Prec) should also be used in evaluating the method.

4.Replay: We gratefully appreciate for your comment. In fact, we calculated the F1_S and Prec of our result. We have performed this 2 criteria in our manuscript. Thanks again fro your valuable comment.

Reviewer 4 Report

The proposed method of using SVM and GBDT to classify EEG activities related to seizures is interesting and timely. While the present manuscript seems to include some promising findings, there are several issues I hope the authors could discuss:

1.     It is unclear how the methods proposed here differ from past classification methods based on similar frameworks (SVM/decision trees). Please clearly elaborate on how the proposed methods will add to the growing body of literation of EEG classification in epilepsy research.

2.     Along the same line, it is unclear what the hypotheses were and why the proposed methods should outperform other methods. Could the authors elaborate on that as well?

3.     It will be helpful to include a result figure where the main findings are illustrated.

4.     Please include more information/description of the exact methods/parameters used in the manuscript. Currently, there is insufficient information for readers to be able to replicate the findings.

5.     What kind of statistical tests were performed aside from computing accuracy/AUC?

6.     There are several typos throughout the manuscript. Please carefully proofread and correct them.

7.     I believe that the readers could benefit from a much more extensive reviews/comparisons of the proposed method to previous studies adopting different methods. For example, here are 3 papers adopting network modeling, dynamical-system based approach, and graph network analysis to probe various aspects of the spatio-temporal dynamics of seizure evolution. How do you think the methods proposed here complement and/or improve these current methods and others?

1) Li et al. (Network Neuroscience, 2018) Using network analysis to localize the epileptogenic zone from invasive EEG recordings in intractable focal epilepsy

2) Lainscsek et al. (Chaos, 2019) Cortical chimera states predict epileptic seizures

3) Sinha et al. (Brain, 2017) Predicting neurosurgical outcomes in focal epilepsy patients

Author Response

Thank you for your decision and constructive comments on my manuscript. We have carefully considered the suggestion of Reviewer and make some changes. We have tried our best to improve and made some changes in the manuscript.

The yellow part that has been revised according to your comments. Revision notes, point-to-point, are given as follows:

1. Comment: It is unclear how the methods proposed here differ from past classification methods based on similar frameworks (SVM/decision trees). Please clearly elaborate on how the proposed methods will add to the growing body of literation of EEG classification in epilepsy research.

1.Replay: We gratefully appreciate for your valuable suggestion. GBDT is a gradient boosting decision tree. GBDT consists of multiple decision trees, and the conclusion of GBDT consists of the superposition of the results of multiple decision trees. GBDT is a strong generalization algorithm. Different from decision trees, GBDT is sensitive to abnormal value, and can reduce the  model bais.

2.Comment: Along the same line, it is unclear what the hypotheses were and why the proposed methods should outperform other methods. Could the authors elaborate on that as well?

2. Replay: We totally understand the reviewer’s concern. We aimed to compare the classification performance between tranditional classifier(SVM) and new classifier(GBDT). We believe that the comparison should be performed in the same variables. Different feature values will affect the classification results. Different parameters of classifiers will also affect the classification results. This paper controlled the variables (feature values and classifier parameters) to cpmpare the classification performance of different classifiers.

And about the question "why the proposed methods should outperform other methods", we don't think anything should be or must be. Our study showed that the classification result of GBDT classifier is better. The other study results may suggest that the classification result of SVM is better. There are many factors affecting the classificationperformance of classifiers. We are still searching for classifiers with good classification performance in any situation.

3. Comment: It will be helpful to include a result figure where the main findings are illustrated.

3. Replay: We gratefully appreciate for your comment. The technology roadmap can help to understand the study. And result figures also can help read. We have performed this 2 figures in our manuscript. Thanks again fro your valuable comment.

4. Comment: Please include more information/description of the exact methods/parameters used in the manuscript. Currently, there is insufficient information for readers to be able to replicate the findings.

4. Replay: We gratefully appreciate for your comment. We have added the details of feature exctration and parameter of classifications in our manuscript. Thanks again fro your valuable comment.

5. Comment: What kind of statistical tests were performed aside from computing accuracy/AUC?

5. Replay: We gratefully appreciate for your comment. F1-score (F1-S), and precision (Prec) can be used in evaluating the method. We have performed this 2 criteria in our manuscript. Thanks again fro your valuable comment.

6. Comment: There are several typos throughout the manuscript. Please carefully proofread and correct them.

6. Replay: We apologize for the poor language of our manuscript. We have now worked on both language and readability and have also involved native English speakers for language corrections. We really hope that the flow and language level have been substantially improved.

7. Comment: I believe that the readers could benefit from a much more extensive reviews/comparisons of the proposed method to previous studies adopting different methods. For example, here are 3 papers adopting network modeling, dynamical-system based approach, and graph network analysis to probe various aspects of the spatio-temporal dynamics of seizure evolution. How do you think the methods proposed here complement and/or improve these current methods and others?

1) Li et al. (Network Neuroscience, 2018) Using network analysis to localize the epileptogenic zone from invasive EEG recordings in intractable focal epilepsy

2) Lainscsek et al. (Chaos, 2019) Cortical chimera states predict epileptic seizures

3) Sinha et al. (Brain, 2017) Predicting neurosurgical outcomes in focal epilepsy patients

7. Replay: We gratefully appreciate for your valuable suggestion. It's important to emphasize the difference between diagnosis and localization in epilepsy diagnosis. The three references mainly focus on the localization of epilepsy through EEG. However, the main purpose of our manuscript is automatic recognition of epileptic EEG in order to qualitative diagnosis. Through the control variables, GBDT, as a new classifier, has a better classification performance than SVM. This technology can reduce the workload of clinical EEG examination. The application of GBDT classifer extends the classification method in the field of EEG automatical analysis. The automatical analysis by GBDT provides a new idea of EEG analysis. In the future, we hope that this technology can help clinicians, reduce the workload of clinical EEG diagnosis, improve the accuracy and efficiency of diagnosis, and benefit patients with epilepsy.

Thank you for your valuable suggestions.

Round 2

Reviewer 1 Report

Authors have improved the manuscript according to reviewers' comments

Reviewer 2 Report

The manuscript can be accepted 

Reviewer 3 Report

The authors of the article have answered most of the questions and the ambiguities have been resolved.

The only important thing left, The comparison of this article with the following references was previously requested from the respected authors, but was not done in the new version of the article.

Comparison with new references such as:.

https://doi.org/10.1016/j.bspc.2021.103417

https://doi.org/10.3390/s21227710

https://doi.org/10.3390/bdcc5040078

Reviewer 4 Report

The authors have thoughtfully addressed all of my previous concerns.